# A Proposed Guideline for Performance of Emergency Surgical Airways in Small Animals: Analysis of Five Unsuccessfully Managed Cannot Intubate, Cannot Oxygenate Cases

**DOI:** 10.3390/vetsci9020039

**Published:** 2022-01-22

**Authors:** Sureiyan Hardjo, Wendy Goodwin, Mark David Haworth, Sarah Leonie Purcell

**Affiliations:** UQVETS Small Animal Hospital, School of Veterinary Science, The University of Queensland, Gatton, QLD 4343, Australia; w.goodwin@uq.edu.au (W.G.); m.haworth@uq.edu.au (M.D.H.); sarah.purcell@uq.edu.au (S.L.P.)

**Keywords:** difficult airway, tracheostomy, hypoxia, intubation, CICO

## Abstract

Objective—To describe three dogs and two cats diagnosed with a cannot intubate, cannot oxygenate (CICO) condition, and discuss the appropriateness and timing of emergency front-of-neck airway access (eFONA). The authors aim to increase awareness of CICO events and effective management strategies, which may result in faster airway access and improve patient outcomes. Case series summary—Three dogs and two cats could not be easily intubated resulting in the inability to deliver oxygen and contributing to their death. Emergency front-of-neck airway access was attempted in three cases, it could not be performed in one when indicated, and it was not considered in managing another. Conclusions—This is the first report of small animals suffering CICO emergencies and tracheostomy attempts without a concurrently secured airway. Cannot intubate, cannot oxygenate events and eFONA attempts were managed sub-optimally in all cases, which likely contributed to the poor outcomes. Rapid diagnosis of CICO and early eFONA using appropriate techniques has the potential to improve the management of difficult airways in small animals.

## 1. Introduction

Difficult airway management (DAM) encompasses a spectrum of clinical scenarios ranging from difficulty delivering oxygen to a patient with a face mask, to difficulty or inability to intubate the trachea [1]. A difficult airway can be due to a physical obstruction or any cause that reduces airway patency. Additionally, it also includes the inability to visualize the glottis or confirm appropriate endotracheal tube placement [1].

The most severe end of the difficult airway spectrum is a cannot intubate, cannot oxygenate (CICO) scenario. This occurs when the trachea cannot be intubated resulting in inadequate oxygen delivery to tissues [2]. In this situation, the only treatment is surgical airway access, termed emergency front of neck airway access (eFONA), which restores oxygenation and ventilation [2]. Cannot intubate, cannot oxygenate events are uncommon in human medicine, accounting for only 0.5% of DAM in hospital emergency departments [3]. To the authors’ knowledge, there are no peer-reviewed publications of small animals with CICO requiring a surgical airway without concurrent airway control. Hence, although the true prevalence of these cases in small animals is unknown, the paucity of reporting suggests it is also uncommon in veterinary medicine. When oxygen supplementation via non-invasive methods fails to improve hypoxemia, and the decision is made to perform orotracheal intubation, the animal is likely already severely compromised. If attempts at intubation are subsequently unsuccessful, the animal is at risk of cardiopulmonary arrest (CPA) due to severe hypoxia. In a clinical situation, it is impossible to predict a timeframe where an animal will arrest from asphyxiation, as experimental studies only assess time to death from a normoxemic baseline in anesthetized animals [4]. Therefore, it is imperative that CICO be rapidly recognized, and efforts made to secure the airway in the shortest time possible.

Veterinarians should be cognisant of the indications for eFONA and be familiar with FONA techniques as immediate action is required for CICO events. The aim of this case series is to describe incidences consistent with CICO, using them as examples to demonstrate how proposed guidelines for diagnosis can prompt rapid recognition and early intervention.

## 2. Case Summaries

### 2.1. Case 1

A female, spayed Burmese cat of unknown age and weight presented for *Ixodes holocyclus* tick paralysis, 24 h following removal of a tick from her left antebrachium. The cat displayed respiratory distress characterized by significant abdominal breathing and respiratory noise localized over the larynx. Sedation was administered; 0.9 mg methadone (estimated 0.3 mg/kg), 0.06 mg acepromazine (estimated 0.02 mg/kg) and 1.5 mg diazepam subcutaneously (SC) (estimated 0.5 mg/kg). Stress was exacerbated by handling as evidenced by two attempts at intravenous (IV) catheterization that resulted in cyanosis. The patient was placed in an oxygen cage for ten minutes before 3 mL of tick antiserum was administered via the intraperitoneal (IP) route. The patient was placed back in oxygen for another hour and a further 2 mL of tick antiserum was administered IP. Intermittent pulse oximetry was performed and readings between 96 and 100% were obtained.

Approximately three hours later, there was excessive noise auscultated over the larynx, which was assessed as laryngeal paralysis. A surgical tracheostomy was attempted but failed as the incision was too caudal and the site obscured by the oxygen mask. Subsequently, numerous attempts at endotracheal intubation were made however correct placement within the glottis could not be visualized. Capnography was unavailable at this clinic to assist in confirming correct placement. An intravenous catheter was then used for percutaneous transtracheal airway access and sutured in position. However, subsequent abdominal distension was indicative of incorrect placement of the catheter with oxygen insufflation into the oesophagus and stomach. The animal became cyanotic and went into cardiopulmonary arrest (CPA), likely due to obstruction of venous return and compression of the thorax. Attempts at resuscitation were unsuccessful. During later post-mortem assessment, the orotracheal tube was visually confirmed to be in the airway.

### 2.2. Case 2

A 14-year-old female spayed Staffordshire bull terrier, weighing 18 kg presented with a 24-h history of dyspnoea. The dog presented with overt inspiratory difficulty consistent with upper airway obstruction. Initial physical examination revealed a temperature of 39.6 °C, heart rate of 120 beats per minute (BPM), wheezing, harsh lung sounds and referred upper airway noise on thoracic auscultation. The owner did not give permission for invasive procedures, including tracheostomy during initial discussions with the attending veterinarian.

Immediate treatment consisted of flow-by oxygen, acepromazine sedation, salbutamol nebulization and dexamethasone. The dog’s dyspnoea worsened, and fly-by oxygen was no longer tolerated. The dog became cyanotic and attempts at placing an intranasal oxygen line resulted in further deterioration. The attending veterinarian elected for intubation and, anaesthetic induction was performed with alfaxalone intravenously, dosed to effect. Intubation was ‘very difficult’ due to excess tissue obstructing the airway. Intubation was eventually performed successfully with a 5.0 mm endotracheal tube however the dog suffered CPA during the DAM and attempts at resuscitation were unsuccessful. Cardiac arrest was likely due to hypoxia and vagal suppression of cardiac function. A laryngeal exam post-mortem revealed a mass associated with the left arytenoid obstructing the airway.

### 2.3. Case 3

A four-year-old female entire dachshund, weighing 6.9 kg, presented to the emergency department with agonal gasping and in cardiac arrest following a dog attack. Chest compressions were commenced immediately, however orotracheal intubation was unsuccessful due to cartilage obstructing passage of the endotracheal tube. A 10 gauge over-the-needle intravenous catheter was passed percutaneously between tracheal cartilages and ventilation commenced with a self-inflating bag device. Epinephrine at 0.04 mg/kg was administered via the intra-tracheal route, using the catheter. An intraosseous catheter was placed in the proximomedial tibia and atropine 0.04 mg/kg was administered. There were no carbon dioxide (CO_2_) readings obtained from mainstream capnometry* at any point during resuscitation. Ventricular fibrillation was noted on electrocardiogram. The owner elected to discontinue resuscitation after ten minutes.

### 2.4. Case 4

A 19-year-old female spayed domestic short hair cat, weighing 3.7 kg, presented for a three-day history of open-mouth breathing and inappetence. The cat was transported to the emergency department and was administered immediate oxygen therapy at the time of presentation. Initial physical examination findings included an inspiratory stridor with an expiratory wheeze, respiratory rate 60 breaths per minute, heart rate 196 BPM, pale pink mucous membranes, and a capillary refill time of 1.5 s. The cat was ambulatory with an orthopneic posture at rest. Salbutamol inhalant was administered as well as butorphanol 0.05 mg/kg SC.

Two hours after admission, the cat’s respiratory character improved, and the respiratory rate had normalized. Thoracic radiography was performed and revealed a soft tissue opacity at the level of the larynx and a bronchial lung pattern. A grand mal seizure occurred whilst in the imaging department and diazepam 0.5 mg/kg IV was effective in terminating the seizure.

An endotracheal tube was placed, and the cat was moved from the imaging department to the treatment area. The tube was displaced during transport and laryngospasm noted. Lignocaine was applied topically to the glottis and re-intubation was attempted with a 2.5 mm endotracheal tube. The patient underwent respiratory arrest, and a percutaneous tracheal catheter was placed using a 10 gauge over-the-needle intravenous catheter. The exact anatomical location of cannula placement was not stated in the clinical record. A self-inflating bag was used to provide ventilation with 100% oxygen. The patient became bradycardic followed by CPA. Cardiopulmonary resuscitation was initiated during which time a tracheostomy was performed. Resuscitation was discontinued following discussion with the owner. The cause of death stated by the attending criticalist was airway obstruction due to laryngospasm.

### 2.5. Case 5

A 4-month-old male entire dachshund, weighing 2.9 kg, with a chronic history of bronchopneumonia presented for an elective laryngeal exam and bronchoalveolar lavage. A serum biochemistry panel, coagulation times and blood smear evaluation were unremarkable at the time of presentation. The dog did not require oxygen therapy before the procedure and was normoxemic during the initial general anaesthesia. Laryngeal function was assessed to be normal under a light plane of alfaxalone anaesthesia IV. The dog was intubated and preoxygenated for 10 min prior to extubation for bronchoscopy. Bronchoscopy revealed mucopurulent material filling the right and left mainstem bronchi. At this point, cyanosis was noted by the anaesthetist and the bronchoscope was removed to perform airway suction. The clinical anaesthetist had difficulty performing orotracheal intubation and airway control could not be confirmed despite multiple attempts.

Seven minutes following the cyanotic event, the patient developed sinus bradycardia, without confirmation of airway control at any point. Cardiopulmonary resuscitation was commenced and performed for six cycles with intubation attempts continued during the first four cycles and intermittent suctioning performed to improve visualization. Electrocardiogram assessment showed pulseless electrical activity at compressor changeover and there were no readings on capnography at any point during CPR. Resuscitation was discontinued after contact with the owner. The cause of death was hypoxia, likely due to airway obstruction by secretions and lack of upper airway control.

## 3. Discussion

The point at which diagnosis of CICO is made during DAM is ill-defined in veterinary medicine, even in popular emergency and critical care texts [5,6]. Hence, early recognition of the event and timing of interventions can be challenging for veterinarians in such a high-stakes scenario. Clinical decision making and errors contribute to most complications during DAM in human medicine [7]. As a result, algorithms, visual aids, and standardized methods are used to objectively assess the patient for CICO and reduce cognitive demand during such crises [2,8]. One report succinctly presented a protocol for diagnosis of CICO based on two categories (procedural and status indications), assessed simultaneously (Appendix A) [9]. This may be more relevant to veterinary medicine, as it does not involve the use of complex airway devices or specialized equipment. An adaptation of this protocol for veterinary use is proposed in Table 1. Any single criterion is diagnostic for CICO, and eFONA should be performed immediately following diagnosis.

Notable additions to the original protocol are 1d and 2c, where eFONA is incorporated into basic life support (BLS) during CPR. Minimal ischemic damage to the myocardium is postulated to occur within the first four minutes following CPA, however, outcomes decline if a perfusing rhythm is not re-established thereafter [10]. This timeframe is used as a guide for procedural indication 1d, inferring that oxygen delivery to the myocardium is markedly reduced while the airway has not been secured. After two minutes of intubation attempts during CPR, one minute is then allotted to set up equipment and establish eFONA, and a further minute of BLS to optimize coronary perfusion pressure if compressions need to be ceased momentarily for any reason [11]. If a patient undergoes CPA as a result of CICO, then eFONA should be performed immediately (2c) to maximize oxygen delivery to an already hypoxic myocardium. An oxygen saturation of below 75% was chosen for indication 1c as veterinary pulse oximeters have poor accuracy during profound hypoxemia and in one human study, the vast majority of deaths during difficult intubations were associated with a similar level of profound hypoxia [12].

The clinical decision-making process for patient 1 could have been improved with the proposed diagnostic criteria for CICO and eFONA. A surgical tracheostomy was attempted before orotracheal intubation in this case. However, orotracheal intubation would typically be attempted first for tick paralysis patients in respiratory distress to address both ventilatory and hypoxic respiratory failure. It is possible the laryngeal noise or obstruction led the veterinarian to believe that intubation would be futile and resulted in direct progression to eFONA. Alternatively, the decision to perform a tracheostomy may have been due to cognitive overload and incorrect decision making during the airway crisis, as intubation should always be attempted first, if feasible. Hence, knowledge of a standardized diagnostic criteria for CICO may have been beneficial in this situation.

Failed intubation attempts, resulting in CPA, were encountered in cases 2 and 5. Both cases were already noted to be cyanotic before airway management commenced, which represents significant and life-threatening hypoxemia (PaO_2_ of 37 mm Hg in the dog with normal haemoglobin content) [13]. Case 2 had a status indication for CICO (2b) as surgical airway requirement was identified as highly probable, though this was precluded due to the owner’s wishes. Once tissue obstruction of the airway was noted and CICO diagnosed (indications 1a and 2a), the animal would likely have benefited from immediate eFONA rather than persisting with the difficult intubation.

Considering induction of anaesthesia occurred prior to the difficult airway in case 5, procedural indications a–c and the first status indication would have been met at some point within the first seven minutes from commencement of airway management until CPA, where eFONA would be of most benefit. Indication 2c was satisfied when the patient arrested during DAM, however, orotracheal intubation attempts persisted during CPR. The commencement of CPR in this case was another time point where eFONA could be of some benefit as it would provide immediate confirmation of airway control, allowing efforts to be concentrated towards resuscitation rather than repeated intubation attempts. In addition to the supplementation of 100% oxygen, this would also allow interpretation of ETCO_2_ to optimize CPR monitoring, as well as allowing a path for more direct suction of the lower airways.

In human medicine, a large study of cases from the American Society of Anesthesiologists legal claims database found persistent attempts at endotracheal intubation in lieu of surgical airway access was associated with death [14]. A smaller Australian report also concluded that numerous deaths could be avoided if not for reluctance to perform eFONA [7]. There is likely a similar reluctance to perform eFONA in veterinary medicine or it may not even be considered an option, such as in case 5. Training and implementation of eFONA procedures in veterinary DAM algorithms may reduce this reluctance to perform eFONA during CICO in small animals. Considering the rapid rate at which a difficult airway crisis can progress to CPA, it is paramount that veterinarians are not only able to diagnose CICO quickly but are prepared to immediately and competently perform an eFONA procedure. The scalpel-bougie cricothyrotomy has been recommended by the British Society of Anaesthetists and the Australian and New Zealand college of Anaesthetists for treatment of CICO [2,15]. In veterinary medicine, a novel scalpel-bougie cricothyrotomy technique has recently been described which may make eFONA more achievable for veterinarians [16]. The technique uses only a scalpel, urinary catheter and endotracheal tube and was found to be significantly faster and easier than a slash tracheostomy [17].

Cases 1, 3 and 4 had percutaneous, cannula eFONA performed. This technique involves placing a needle or intravenous catheter in to the airway to provide short term oxygen therapy to animals with upper airway obstructions [5]. In human clinical practice, this technique has a very high failure rate [18] and is not the primary option recommended for CICO as surgical techniques are superior [2]. An experimental study in pigs found only a 20% success rate at delivering rescue oxygenation, citing frequent kinking or displacement of the catheter as reasons for failure [19]. This may explain the outcome of both cases that underwent CPA with a cannula tracheostomy in place (cases 1 and 3). With regards to case 4, a 10-gauge catheter with a 2.4 mm internal diameter was used for percutaneous airway access, which could be considered a reasonable size tube for a small cat. In a canine model, oxygenation and ventilation was maintained indefinitely in medium-sized dogs (26 ± 2 kg) with airway obstructions using percutaneous 12-gauge intravenous catheters [20]. The reason for CPA after successful cannula airway access in this case is unclear but it could possibly be due to cannula displacement or kinking.

Cannula airway access is also not ideal during CPR compared with endotracheal intubation due to lack of a tube cuff sealing the airway, which is likely one of the reasons case 4 had the cannula tracheostomy immediately converted to a tube tracheostomy. It is impossible to provide effective ventilation or oxygenation against the force of a chest compression without adequate sealing of the airway [21]. Another reason favouring tubes over cannulas during CPR is improved monitoring. It should be noted that eFONA may not have changed the outcome in case 3, which could have died from multitude reasons, including internal haemorrhage, chest trauma or myocardial contusion. Nevertheless, with case 3, end tidal CO_2_ may not have been detected from the in-circuit capnometer as exhaled air would take the path of least resistance around the catheter and through the larynx, making monitoring difficult. These reasons favour eFONA using endotracheal tubes over cannula techniques during CPR.

Two cases had eFONA performed during CPA (3 and 4) to address airway obstructions, therefore, the prognosis would have been very poor even before the eFONA attempt. If CPA is a result of myocardial hypoxia secondary to CICO, immediate oxygen supplementation via eFONA during CPR is vital to establish return of spontaneous circulation (ROSC). There are sentiments in human medicine that chest compressions should even be ceased to ensure timely airway control by cricothyrotomy if CPA is a result of CICO [22]. Further, one human study demonstrated early invasive airway access had the potential to improve the outcome of CPR for patients suffering from in-hospital CPA [23]. Hence, there should be some emphasis placed on eFONA during BLS in veterinary CPR for difficult airways in order to optimize results in an already grave situation.

There are no veterinary studies or guidelines addressing the diagnosis of CICO in small animals. Hence, when the animal’s status is unclear, the emphasis must be placed on rapid recognition and intervention based on the inability to rule out the diagnosis. If an animal undergoes CPA, the overall rate of ROSC remains relatively low for CPR (44% in dogs and 55% in cats) [24]. If CICO is the primary condition, eFONA using cricothyrotomy has the potential to be immediately life-saving, with this procedure having a 100% success rate for some techniques in humans [25]. Therefore, the consequences for performing potentially unnecessary eFONA after satisfying at least one of the CICO criteria are not as severe as failing to perform it when required.

It should be noted that eFONA is the last resort during DAM and basic, non-invasive methods of stabilization must be performed first [26], which is beyond the scope of this discussion. Other advanced airway management techniques that may assist orotracheal intubation, which were not demonstrated in the case series include the bougie-assisted airway in a rapid sequence intubation protocol, endoscope assisted intubation and retrograde guidewire assisted intubation. Regardless of the method chosen to attempt intubation, the proposed diagnostic criteria for CICO within this manuscript can serve as specific and objective critical limits for a veterinarian to consider performing eFONA.

## 4. Conclusions

In conclusion, this report presents various examples of CICO in small animals, that all succumbed to their primary illness. It is probable that these animals would have benefited from timelier and more appropriate eFONA. Rapid diagnosis of CICO can be improved in veterinary medicine, where it should be recognized that airway obstruction is not the only indication for eFONA. Difficult airway management can potentially be assisted with visual aids and algorithms, similar to those used in CPR. Veterinarians more likely to encounter CICO, such as emergency and anaesthesiology clinicians, should regularly prepare for these situations in their training to allow for early diagnosis of CICO and improve confidence in performing eFONA.

## Figures and Tables

**Table 1 vetsci-09-00039-t001:** Proposed guidelines for diagnosis of cannot intubate, cannot oxygenate and indications for emergency front of neck airway access in small animals during difficult airway management.

1—Procedural Indications	2—Status Indications: Supercedes Procedural Indications
Three unsuccessful attempts at orotracheal intubation by a suitably qualified veterinarian 10 min elapsed since anaesthetic induction without confirmed airway intubation Oxygen saturation falls below 75% (or the patient becomes cyanotic) at any point after the first or second intubation attempt 2 min from the commencement of CPR without confirmed airway (non-respiratory causes of arrest)	Rapid desaturation during orotracheal intubation attempts Criticalist, anaesthetist or other suitably qualified veterinarian decides further attempts at orotracheal intubation would be futile Hypoxemic patient arrests during management of difficult airway

Note, an airway obstruction, mass or foreign body is not the only reason to diagnose CICO and perform eFONA.

## Data Availability

Not applicable.

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
