# Peer review of "A Proposed Guideline for Performance of Emergency Surgical Airways in Small Animals: Analysis of Five Unsuccessfully Managed Cannot Intubate, Cannot Oxygenate Cases"

_vetsci, 2022, doi:10.3390/vetsci9020039_

Round 1
Reviewer 1 Report
The establishment of guidelines for the diagnosis of CICO will be very useful for the management of theses cases. The proposed guidelines will have to be tested on a large cohort of animals to ensure their validity. Also, it will be interesting to have guidelines for the specific management of these cases.
Specific comments:
Line 2: I said Preliminary guideline rather that A guideline
Line 68: How do you explained that the trachea could not be exposed for the tracheostomy?
Line 268: That should be interesting to have a recommendation on the different techniques of eFONA to use. Is it possible to recommend one technique rather than another (perhaps in function of the clinical presentation)? Maybe, you can list the different techniques in a Table.
Author Response
"Please see the attachment."

Reviewer 2 Report
Dear Authors,
Thank you very much for addressing this interesting topic. Difficult airway management is observed in clinical practice, and the worst scenario of CICO can be life-threatening.
My concerns refer to the case selected. Some critical information is missing, and the description of the clinical procedures do not highlight the relevance of the eFONA technique. The definition of CICO is mentioned in the discussion, but it should be moved in the introduction. This change will emphasise the issue and contextualize the cases reported.
Below I added further comments:
Lines 37-40 “…therefore it is also likely to be rare in veterinary medicine. There were seven cases found on a record search of three clinics over a span of 19 years, none of which survived. Five cases had enough information to present in this report.” This is a strong statement. Compared to human medicine, accidents/complications in veterinary practice are not reported systematically. Indeed, the authors state that the five documented cases are the most complete of a series of different episodes observed in 3 clinics over 19 years. I suggest removing this statement.
The authors do not mention the weight of the animals in the description of the cases.
The rationale of the administered drug at admission (e.g. salbutamol) is unclear. Indeed, the authors do not mention the possible differential diagnosis of dyspnoea.
In cases 1, 2, 4 and 5, there is no hypothesis of the causes that have led to cardiac arrest, despite properly secured airways (cases 1, 2 and 4).
The authors do not describe the technique eFONA. At which level of the trachea was the intravenous catheter placed?
Line 86 please, change fly-by to flow-by
The authors do not provide information on the inclusion criteria for considering the patient at risk to develop CICO. In the introduction, the definition of CICO, proposed by the authors based on the study from Dillon et al. 2013, should be included.
In the discussion section, the authors should discuss why eFONA did not improve the clinical outcome despite the correct positioning of the catheter (case 1). Performing a proper technique does not seem to guarantee an improvement. Other factors such as sedation/anaesthesia or previous medical condition could represent factors that may negatively affect the outcome.
In the discussion, the authors do not mention different approaches to secure airways. An endoscope (a portable one) or a guidewire or a stylet could facilitate endotracheal tube insertion. Moreover, the use of local anaesthetic (lidocaine splash or xylocaine nebulizer) to desensitize the larynx can be considered in such a difficult case.
Lines 187-188 I would mention in the sentence that cyanosis indicates a PaO2 of 37% but in dogs with a haemoglobin concentration within physiological limits. In the cases presented, there is no reference to the results of haematological exams.
